# ALGORITHM DESIGN FOR LEARNED ALGORITHMS

## ABSTRACT

Neural networks can learn known algorithms from data even when only trained with input/output pairs and no supervision over the intermediate steps. This means that with labelled examples, networks can potentially learn new algorithmic approaches. Engineers designing new algorithms are often faced with trade-offs such as efficiency versus accuracy or generality versus specificity. In this work, we show that the same controls exist when learning algorithms from data and we explore how model hyperparameters control the accuracy, efficiency, and generality of the resulting algorithm. Our analysis covers learned approaches to computing prefix sums and solving mazes; these domains have existing fast and accurate solvers so they serve as a great test-bed for our analysis. Finally, we extend this analysis to learning algorithms for constraint satisfiability – an NP-Hard problem.

## 1 INTRODUCTION

Recurrent neural networks designed to learn and perform algorithms (called Deep Thinking Networks or DT-Nets) show promise on simple problems for which we have classical alternatives, however utilizing them for novel algorithm synthesis remains an open problem. In order to employ these models in domains where exact solvers are slow, it is critical to study their behavior and uncover their ability to model fast approximate solvers.

In this work we show that a user training a network has some control over the resulting algorithm and we use visualization to analyze these algorithms post-hoc. We pinpoint the hyperparameters that act as knobs to turn to fluidly shift from fast and approximate algorithms to slow and accurate ones. Much of these results come from computing prefix sums and solving mazes, relatively simple problems that provide a rich space for experimentation and comparison to known classical algorithms. These problems are in the complexity class P and there are lots of classical algorithms to compute solutions in polynomial time. However, algorithms for approximating solutions to NP-Hard problems such as constraint satisfiability (SAT) are often much more complicated and scale poorly to large problem sizes. We take all the lessons we learn from simpler problems and demonstrate that they scale to NP-Hard problems like SAT, using the same framework. Importantly, this shows that recurrent neural networks are capable of stepping outside the regime of solving problems for which practitioners have many known options and into the space where discovering new algorithms is of great value.

### 1.1 RELATED WORK

Drawing on prior work on extrapolating from easy to hard with recurrent neural networks, we show the promise in using neural networks to find new algorithms and discuss the control we have over the algorithms these networks learn. Neural algorithmic reasoning is commonly used to refer to for the process of training a neural network to simulate an algorithm or reason about a task in a scalable fashion. To promote neural algorithmic reasoning, a recurrent unit is often employed that simulates one step in the reasoning process.

Much existing work focuses on algorithmic reasoning by graph neural networks (GNNs) due to the expansive capabilities to operate on generic forms of data with graph structure (Veličković et al., 2020; Xhonneux et al., 2021; Ibarz et al., 2022; Bevilacqua et al., 2023). In fact one benchmark dataset lifts 30 algorithms from a classical algorithm analysis text book and offers graph-based representations of the problems and algorithmic solutions (Veličković et al., 2022). Several papers on

GNNs that can solve these problems focus on building neural networks that execute the particular steps of a known algorithm, like breadth-first search Veličković et al. (2020); Xhonneux et al. (2021); Ibarz et al. (2022); Bevilacqua et al. (2023). A unifying theme among these works is that networks are taught a specific algorithm and they are often trained with hints or supervision over the intermediate steps.

Our focus is on learning algorithmic behavior from data without *a-priori* choosing a known algorithm to emulate (e.g. Bansal et al., 2022). This in turn, allows for applications of these networks to a more diverse domain as intermediate steps need not be known. While Ibarz et al. (2022) remove teacher forcing from their recurrent GNN training, they do still attempt to emulate steps of known classical algorithms. With this distinction in mind, we build directly on work that finds algorithms from problem/solution pairs without intermediate information or supervision.

In our work, the particular neural network architecture we study is at the center of prior work on exactly this type of algorithm synthesis (Schwarzschild et al., 2021b; Bansal et al., 2022; McLeish & Tran-Thanh, 2023). Schwarzschild et al. (2021b) introduce a generic model for adaptive compute models using recurrent networks and coin this family of models Deep Thinking networks (DT-Nets). In follow up work, Bansal et al. (2022) improve on the model architecture and the training routines showing impressive generalization from small/easy training sets to hard samples at test time. In particular, Bansal et al. (2022) also define the notion of *overthinking*, a sharp decline in accuracy after a problem has been solved and a behavior we show the user has control over with hyperparameter tuning. These works focus on benchmark datasets for studying logical extrapolation or algorithmic reasoning that include computing prefix sums and solving mazes – tasks we use in our experiments below (Schwarzschild et al., 2021a).

We take the ideas in those works and apply them to SAT solvers. The broad class of constraint satisfiability problems has two main variations: a decision problem and an optimization problem. In both instances we are given a Boolean formula in conjunctive normal form as input. For decision problems, we ask if it is possible to satisfy this formula, i.e. if there exists an assignment for all the variables that make the formula true. For the optimization problem we ask for an actual assignment of each variable that satisfies this condition. SAT problems with clauses in the given formula with length longer than two are NP-Hard. In this work we focus on the optimization problem, thus addressing how learned algorithms can tackle NP-Hard problems.

In prior work, Wang et al. (2019a) create a differentiable module for solving the semidefinite program associated with the MAXSAT problem, learning this structure in an unsupervised fashion. MAXSAT is a version of the optimization SAT problem where we aim to satisfy as many clauses in the formula as possible even if we fail to satisfy them all. Wang et al. (2019a) also solve Sudoku puzzles by using a convolutional neural network combined with their MAXSAT network. Selsam (2019) solves the decision SAT problem for formulas which are generated by adding clauses to a formula until it becomes unsatisfiable. This technique uses graph neural networks for this classification task processing both satisfiable and unsatisfiable examples. By inspecting the clusters formed by the embedding of the literals or using principle component analysis, the optimization SAT problem can then be solved using the same network, showing extremely high in and out-of-distribution accuracy. In other work, Wang et al. (2019b) use feedforward convolutional neural networks for the optimization SAT problem. They use their network in combination with an off-the-shelf solver in order to satisfy the formula. The authors compare with many concurrent state-of-the-art models and show strong results in these comparisons. This work is still currently being expanded (e.g. Wang et al., 2021), where models inspired by Selsam (2019) are being adapted and combined with off-the-shelf solvers to improve the abilities of neural networks to solve SAT problems. In this work, we use the end-to-end DT-Net approach to solve instances of the optimization SAT problem and control the resulting reasoning scheme the model learns.

## 1.2 BACKGROUND

We use the architecture and training routines as well as the language used in prior Deep Thinking work (Schwarzschild et al., 2021b; Bansal et al., 2022). To begin our discussion of these models, we briefly review the model architecture. DT-Nets are fully convolutional models that operate best for problem formulations where the input and output have the same spacial dimension. They are based on ResNets (He et al., 2016), but in DT-Nets a single residual block is iterated several times

rather than having distinct layers. These iterative models are trained with a maximum number of recurrences – we refer to this hyperparameter as $\mu$. The effect of $\mu$ has not been explored in depth in prior work, but we find that it has a lot of control on the final model. At test-time, the recurrent block in these models can be called any number of times providing us with a notion of "test iterations" that counts how many recurrent steps the model performs.

In evaluating these models, Bansal et al. (2022) study a phenomenon called *overthinking*. Overthinking is a decline in accuracy that occurs with too many test iterations as a result of solution degradation after a problem has been solved.

Prior work makes clear that DT-Nets are capable of learning scalable algorithmic processes that can solve problems of arbitrary size when trained only on small examples. This sets the stage for our main inquiry into whether we recognize these algorithms and what in the pipeline pushes these models to learn one algorithm over another.

## 2    What Algorithms Do We Learn?

When we say that neural networks learn algorithms we mean that they extract a scalable process from data and they can generalize from easy to hard. But what do they really learn? To look closer at this question we probe these models using visualizations. That is, we actually observe the learned process by showing solutions as they evolve with iterations of the recurrent block.

### 2.1    Maze Solving: A Visual Case Study

One technique for which recurrent networks are well suited is to look at the iterative outputs to glean something about the process they are carrying out. To make the learning process more clear we are able to set $\mu$ (the maximum recurrent iterations) as a hyperparameter of the network and observe the progress of the solution as a function of the current iteration. Thus, we can run for one iteration at a time and look at the solution at that moment before we run the next iteration.

**Maze solving is done by dead-end-filling.** In Figure 1, we show a representative example where we can see that the solution evolves as dead ends are back filled and removed from the solution by looking at the output in the intermediate steps by the model trained with $\mu = 30$. We further show in Figures 17 and 18 (in Appendix A.2) that this behavior is consistent across maze examples and that this behavior is consistent across different models trained with the same hyperparameters. This algorithm is a known process for solving path finding tasks that works only when the underlying graph is a tree. Note, that the Mazes Datasets from Schwarzschild et al. (2021a) contain only tree-based mazes where there are no cycles, i.e. there is a unique path connecting any two cells in the maze.

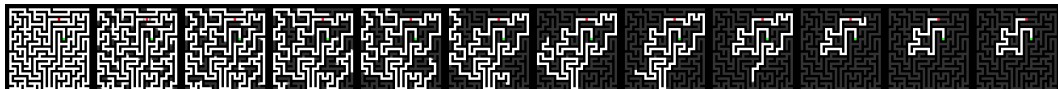

Figure 1: Maze solving DT-Nets learn dead-end filling, a known algorithm that works to solve mazes without cycles in the path structure. The first image is the maze to solve (input) and subsequent mazes are the solutions (outputs) at 2, 4, 8, 16, 32, 50, 70, 100, 140, 160, and 220 iterations, respectively.

**DT-Nets do dead-end filling with an error correcting twist.** While it is clear that DT-Nets perform dead-end filling when solving a maze from scratch, we can shift our attention to how they respond to swapping the puzzle after some number of iterations have already taken place. Bansal et al. (2022) explore exactly this behavior quantitatively, showing that these networks can recover the solution to the new maze. But in this experiment we are focused on what process the networks employ to do this.

In Figure 2, we show this error correction process by starting at the iteration where the output matches the correct solution to one maze, but we move the red and green dot so that solution is no longer correct. In this case, as the model iterates, it's able to connect the new start and end locations with a more-or-less head on approach. We do not see exclusively dead-end filling, rather

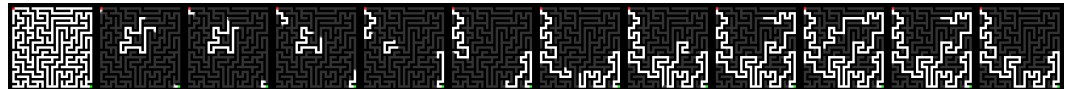

Figure 2: Starting from the end point of Figure 1, we move the start and end points of the maze to different positions and see the model perform a search to reconnect the endpoints. The first image in this figure is the maze with modified start and end points and the subsequent mazes are the solutions (outputs) at 2, 4, 8, 16, 32, 50, 70, 100, 140, 160, and 220 iterations, respectively.

some more intelligent process is at play. Hence we conclude that the learned maze-solver is not just a dead-ending filling algorithm, but is smarter as it can switch strategy based on the changes in the maze.

## 2.2 WHEN WE DO NOT LEARN DEAD-END-FILLING

In this subsection, we conduct experiments to investigate if the Deep Thinking maze solvers always choose a dead-end filling algorithm to find the path between any two points. For this we constrain the models during training to use a smaller value of $\mu$. Specifically, we train models with $\mu \in \{15, 20, 30\}$. In Figure 3, we can see that the models trained with $\mu$ equal to 15 or 20 learn a search-like algorithm that is quite different from the algorithm learned when $\mu = 30$. Furthermore, we also show in Figures 19 and 18 that for different trials training models with $\mu$ equal to 15 or 20, some learn the search algorithm, while others learn the dead-end filling algorithm. Thus, we see that by simply changing the maximum number of iterations allowed during training, one can have a model that learns different algorithms.

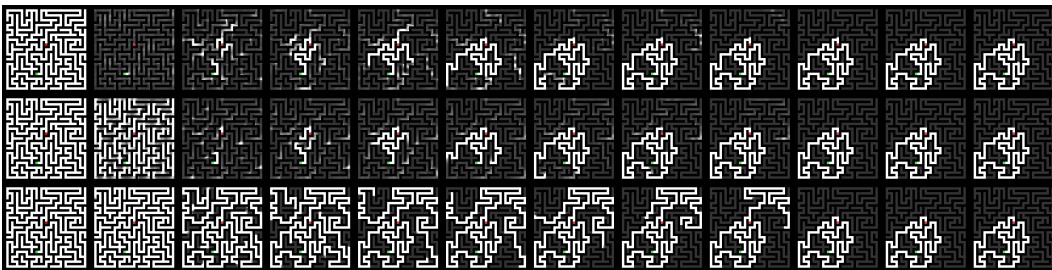

Figure 3: We present the outputs of the intermediate steps for the three different models. The top row shows output from a model trained with $\mu = 15$. The middle row reflects $\mu = 20$, and the bottom row shows results from training with $\mu = 30$. We can see that the models execute different maze-solving algorithms. The first image in this figure is the maze with modified start and end points and the subsequent mazes are the solutions (outputs) at 1, 4, 8, 16, 24, 32, 40, 50, 100, 150, and 200 iterations, respectively.

Using mazes as a case study, we confirm that neural networks indeed manage to find algorithms while training on input/output pairs alone. Remarkably, the algorithms crafted by these models exhibit a degree of interpretability and can be easily altered by adjusting the maximum number of recurrences used during training. The interpretability of the resulting algorithms is surprising as interpretability is often difficult with neural networks. This new-found clarity provides a glimmer of hope; as Deep Thinking models find application in real-world scenarios, there is potential for unraveling the workings of their solutions.

## 3 THE IMPACT OF $\mu$

One major trade-off of algorithm design is accuracy and efficiency. While it is clear that DT-Nets learn algorithms and that we have some tools to better understand what those algorithms are like, the next big question is can we make them faster?

Since the maximum number of recurrences during training, $\mu$, is a hyperparameter, we explore the impact of limiting that number.

## 3.1 FASTER PREFIX SUM ALGORITHMS

Reducing the number of training iterations rewards the model for learning faster reasoning schemes, as the loss is calculated on a reduced number of iterations of the recurrent module. This has an impact on the accuracy of any model, as solving any problem over a longer period of time is naturally easier.

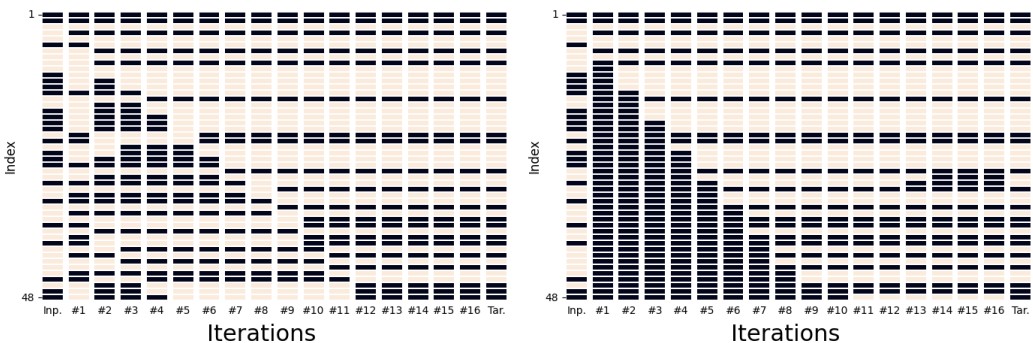

Figure 4: **Left:** Prefix Sums model trained with $\mu = 30$. **Right:** Prefix Sums Prefix Sums model trained with $\mu = 6$

Figure 4 shows a representative example comparing the solution evolution from two networks – one trained with $\mu = 30$ during training and one trained with only $\mu = 6$. On the left of Figure 4, we see that a model trained with a higher number of iterations during training appears to work sequentially with random assignments for bits it has not yet resolved. On the right of Figure 4, we see that the model trained with a fewer number of maximum iterations appears to assign all bits value zero, then work sequentially to fix these bits to the correct value. Thus demonstrating that with a different value for $\mu$, a different algorithm and reasoning scheme is learned.

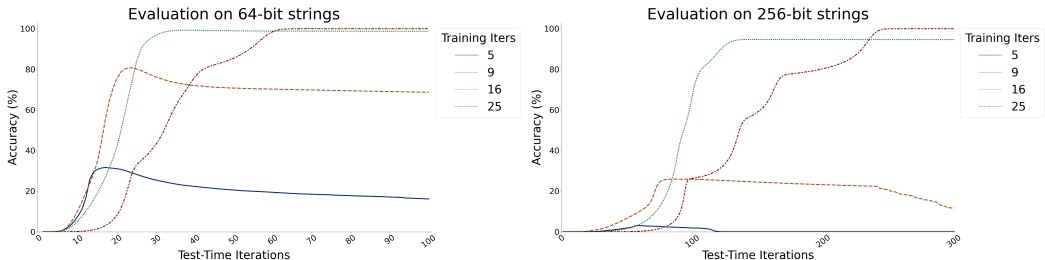

Figure 5: Prefix Sums models trained with varying values of $\mu$, tested on: **left:** 64 bit data, **right:** 256 bit data

We see a trade-off between accuracy and speed for prefix sums model, our most simplistic test bed, in Figure 5. We see that models trained with a larger value of $\mu$ achieve higher accuracy slower than models trained with a smaller $\mu$. So, reducing the value of $\mu$ can allow us to position our model within this speed and accuracy trade-off to achieve the desired properties.

We see an interesting linear phenomenon in left of Figure 6, plotting the average time to solve a prefix sums problem with models trained with varying values of $\mu$, with a larger constant factor in the near linear term for models with larger values of $\mu$. Note that in Figure 6 we are only counting problems the model gets correct within the maximum number of testing iterations.

### 3.1.1 OVERTHINKING

The primary knob that dictates the trade-off between accuracy and speed of learned algorithms is $\mu$, the maximum number of iterations during training. Taking a closer look at this relationship, we see a phase shift from non-convergent imperfect algorithms that overthink to algorithms that can solve

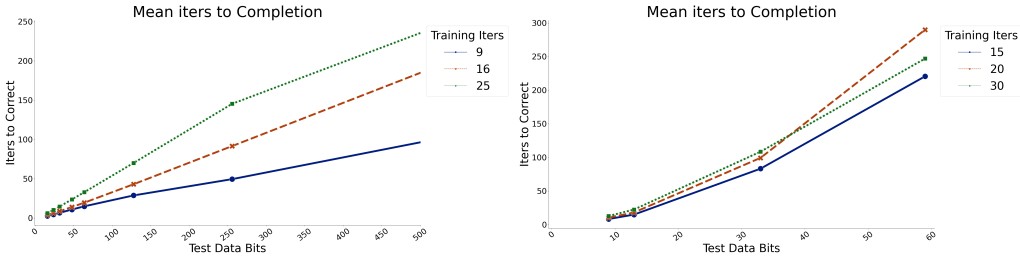

Figure 6: **Left:** Mean number of testing iterations to solve Prefix Sums problems on all sizes available in the dataset, varying over the value of $\mu$. **Right:** Mean number of testing iterations to solve Maze problems on all sizes available in the dataset, varying over the value of $\mu$.

all test samples and do not overthink. We isolate the particular values of $\mu$ where this phase shift takes place and highlight that we have fine-grained control over the speed/accuracy in some cases.

From Figure 5, it is clear that there is a direct correlation between the value of $\mu$ and the learned algorithm's ability to generalize to harder problems. Additionally, algorithms that have been trained with a sufficiently large $\mu$ can solve problems of arbitrary size and algorithms learned by models with smaller $\mu$ values suffer from overthinking. This effect can be observed in Figure 7.

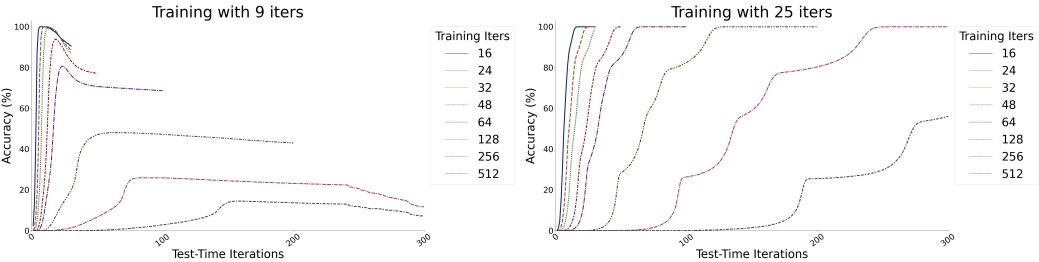

Figure 7: Prefix Sums models trained with $\mu$ equal to 9 and to 25 (left and right, respectively) tested across prefix sum inputs of varying length. Note, $\mu = 9$ is not enough to reliably genearlize to harder problems, but when $\mu = 25$ models can solve problems of arbitrary size.

## 3.2 FASTER MAZE SOLVERS

Moving to a more complex domain, we analyze DT-Nets in the setting of solving mazes. In Figure 8, we see that training with a smaller value of $\mu$, leads to models which achieve higher accuracy more quickly but also overthink. These models trained with fewer recurrences do not have the fixed-point behavior that models trained with more recurrences show.

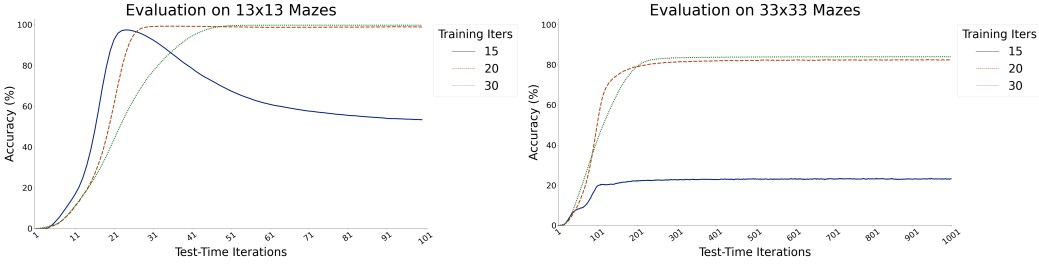

Figure 8: Maze models trained with varying value of $\mu$ tested on: **left:** 13x13 data, **right:** 33x33 data

The right side of Figure 6 shows a similar story when we measure the average time to reach a solution for mazes. Note, the quadratic scaling here reflects the iterations to completion as function

of the maze side length, so this shows linear scaling with respect to the number of cells in a puzzle, For example, when a maze is labelled size 13 on the x-axis, there are $13 \times 13 = 169$ pixels in the image.

While these results show that one can learn a faster algorithm by turning the knob on the number of recurrences during training, a natural question arises as to why anyone would use approximate solvers for computing prefix sums or solving mazes. However, we feel that these domains serve as a sandbox in which we can experiment, visualize, and reason about how the hyperparameters affect the resulting algorithm the DT-Nets learn. This is particularly important to get a handle on if one is interested in utilizing DT-Nets in domains where analysis and comparisons to known algorithms would be challenging.

## 4    SAT PROBLEMS WITH DT-NETS

To enter the NP-Hard class, we shift our attention to SAT problems. In this space we have the exact setting where fast approximation is critical, as we have no known polynomial time exact solvers. Since the accuracy/efficiency trade-off is a constant battle for NP-Hard problems, algorithm synthesis with a knob to turn is valuable.

In this section, we train DT-Nets to solve SAT problems and analyze how the lessons from Section 3 apply in this complex problem domain.

### 4.1    PROBLEM SETTING

The constraint satisfiability problem (SAT) is a problem where given a Boolean formula in conjunctive normal form, we attempt to assign each variable a value of true or false in such a way that all clauses within the formula are satisfied, hence the formula evaluates to true. When clauses are allowed to have length greater than two this problem is NP-Hard. SAT can be represented as an image, with columns representing literals and rows clauses. We can then fix the colour of the corresponding pixel dependent on whether a variable is in a clause, with padding to allow for changes in the size of the input within a batch, an example is shown in Figure 9. This allows us to apply the Deep Thinking convolutional architecture to the SAT problem, with three input channels: present in clause, not present in clause and not present in the entire formula.

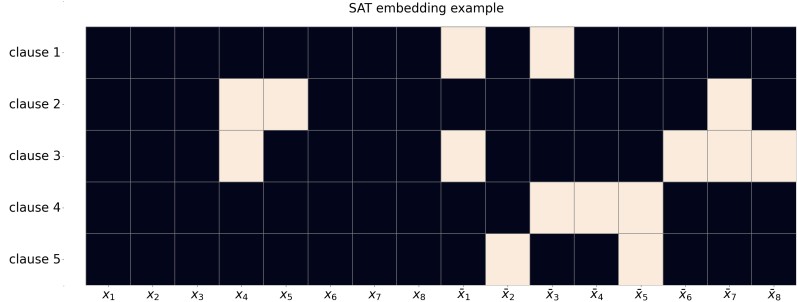

Figure 9: Embedding of $(\overline{x_1} \vee \overline{x_3}) \wedge (x_4 \vee x_5 \vee \overline{x_7}) \wedge (x_4 \vee \overline{x_1} \vee \overline{x_6} \vee \overline{x_7} \vee \overline{x_8}) \wedge (\overline{x_3} \vee \overline{x_4} \vee \overline{x_5}) \wedge (\overline{x_2} \vee \overline{x_5})$

To generate the data we use a similar approach to Selsam (2019). We sample from a uniform distribution twice, once to fix the number of clauses in the formula and once to fix the maximum number of non-complementary variables formula. We then randomly (uniformly) pick a number of non-complementary variables between one and the maximum number for each clause. We then select these literals randomly with removal from the set of all literals for each clause, choosing them to be positive with probability one half. We then use MiniSAT (Een & Sorensson, 2005) to check that the formula is satisfiable before adding it to the dataset.

We train and validate on data where the number of clauses is chosen from $U(5, 10)$ and the number of non-complementary literals in the formula is chosen from $U(5, 10)$. We test on data where the number of clauses is chosen from $U(10, 14)$ and the number of non-complementary literals in the formula is chosen from $U(10, 14)$ and data with 11 clauses and 11 non-complementary literals in

the formula. The size of the training data is 10,000 and all other test sets are 1,000 samples from a possible dataset of 10,000 samples.

## 4.2 RESULTS

We show in Figure 10 that the same controls as seen in Sections 3.1 and 3.2 can be used to control SAT models. We compare three models, one trained with $\mu = 20$, one with $\mu = 30$, and one with $\mu$ equal to 40 , over 1000 SAT instances. We see the same overall trend, that a smaller $\mu$ creates a model that can reason quicker but will overthink as opposed to models trained with a larger $\mu$ value, which take longer to reach a solution but can then maintain it. In Figure 10, we also see models can learn fixed point algorithms (no overthinking) when $\mu \geq 30$. This is in contrast to a model trained for fewer epochs (20 as opposed to 30), for which the value of $\mu$ controls accuracy more directly; this can be seen in Figure 12. A compilation of the graphs shown in Figure 10 can also be seen in Section A.1, for further comparison. Representative examples of the outputs of multiple testing iterations of the model can be seen in Figures 11 and 15.

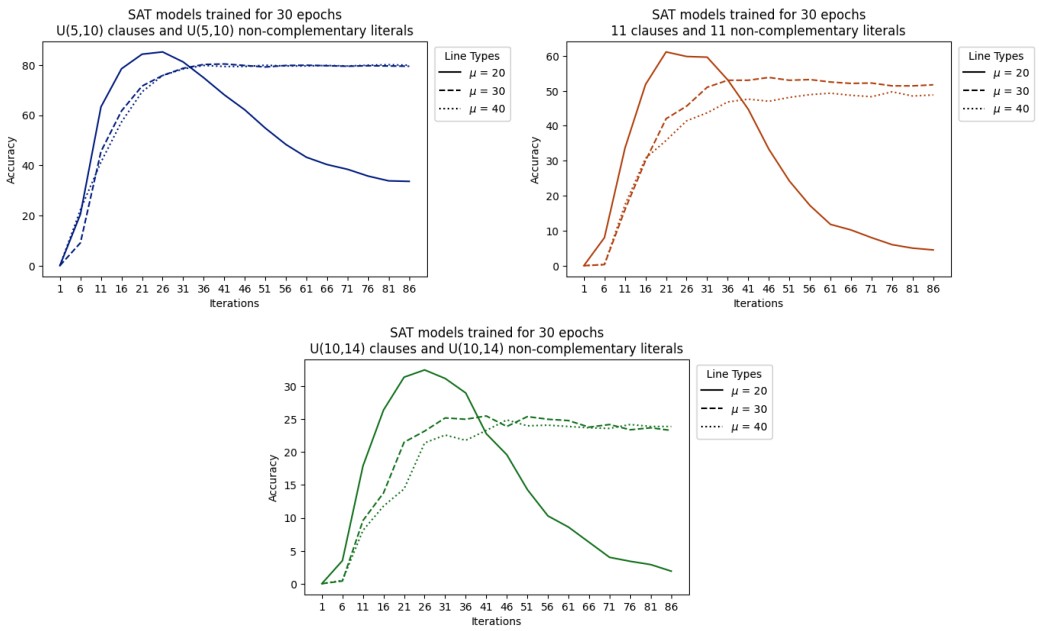

Figure 10: SAT models trained for 30 epochs, trained with varying values of $\mu$

It is no surprise that DT-Nets extrapolate to different types of Boolean formulas but with room to improve. We demonstrate that the same algorithmic knobs which apply to polynomial time solvable problems can be extended to the SAT problem, allowing for wider exploration of the neural algorithmic space for NP-Hard problems.

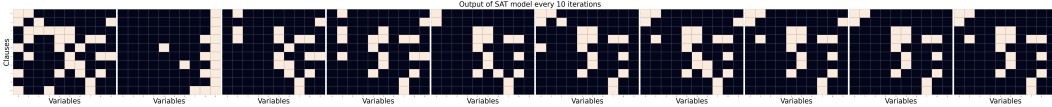

Figure 11: Representative SAT model output every 10 iterations when solving a random training data sample, during testing. The input is shown on the far left.

## 5 DISCUSSION

We demonstrate the ability to control the reasoning capabilities learned by Deep Thinking models on prefix sums and mazes – domains for which we have polynomial time classical algorithms. We also

show we can turn knobs on hyperparameters, most prominently on the number of recurrences used during training, $\mu$, that in turn change the properties of the resulting algorithm. We show visually for mazes and prefix sums, that we can identify familiar patterns in the learned reasoning scheme and that it changes as we vary $\mu$.

We then extend to the domain of constraint satisfiability, a problem for which no polynomial time algorithm currently exists. Showing, within the same Deep Thinking framework, that we can maintain the same control over accuracy and speed.

It is well known that each SAT problem instance falls within one of the following three regimes: (i) easy to solve (i.e., we can find an optimal solution in P); (ii) notoriously difficult to solve: the problem instance is a provably NP-Hard instance; and (iii) borderline regime, where it is not clear whether there is an easy solution for it. Finding good heuristics to offer reasonable running time when solving the SAT problem may be optimal for classical computers. If this is the case, in the future, using Deep Thinking networks to offer insights into how to develop heuristics and novel reasoning ideas for the SAT problem, or even be used as a heuristic itself, may be of great value when solving some of the hardest problems in computer science.

Our results on SAT solving raise exciting questions and prompt future work on applying neural algorithms to NP-Hard problems. For example, we observe that the models that overthink have higher peak accuracy and that peak occurs with fewer test-time iterations. Another observation is that the best accuracy across all the SAT solvers we examine is far from 100%.

As is the case across classical algorithm analysis and neural algorithm synthesis, there is room to improve the performance and behavior of our best methods when it comes to NP-Hard problems. Perhaps learning new algorithms from data will help us better understand and overcome these challenges.

## 6    REPRODUCIBILITY

The majority of our experiments can be carried out using only code made available with prior work (Bansal et al., 2022). We develop new training and testing scripts for SAT models and we provide these scripts in the supplementary material. For those experiments, we generate labelled data according to the methods described in Section 4.1. We provide 10,000 data samples for all the datasets we use, as well as data with 12 clauses and 12 non-complementary variables; 13 clauses and 13 non-complementary variables; $U(10, 20)$ clauses and $U(5, 10)$ non-complementary variables and a validation set with the same parameters as the training data set. Together with our appendix, the supplementary material makes reproducing our results accessible and easy.

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

# A APPENDIX

## A.1 SAT

In Figure 12, we see a similar plots to Figure 10 but for models trained for 20 epochs. We see for these models, the value of $\mu$ is more important than it was for models trained for 30 epochs when extrapolating to harder data.

Figures 13 and 14 are a compilation of Figures 10 and 12 respectively, to allow the reader to compare the models over varying data sizes.

## A.2 MAZES

## A.3 HYPERPARAMETERS

All models are trained with an Adam optimizer and the Deep Thinking framework which was released on GitHub (Bansal et al., 2022).

## A.4 PREFIX SUMS VISUALIZATIONS

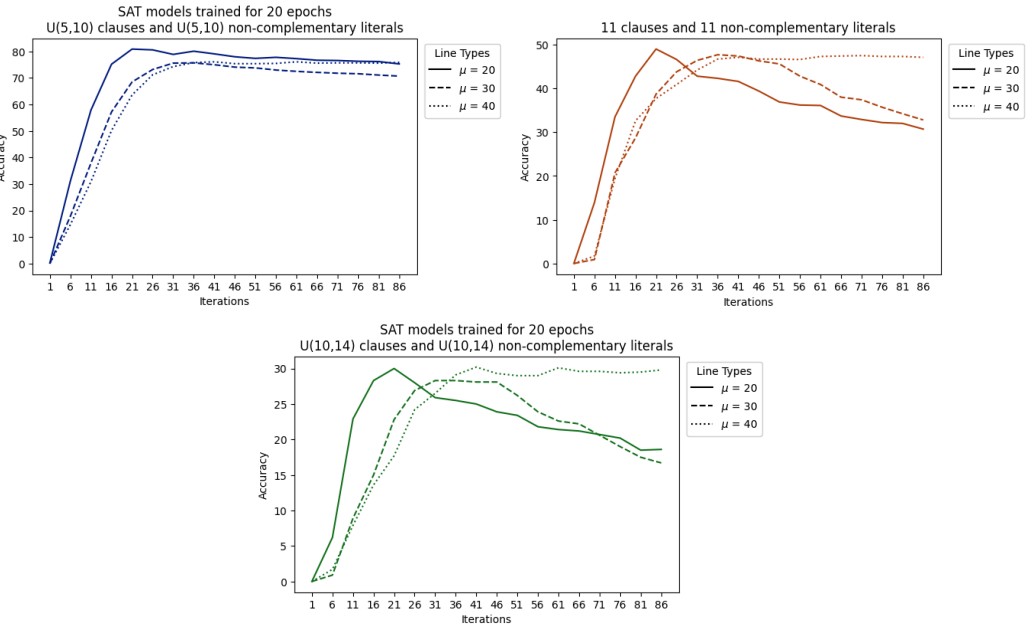

Figure 12: SAT models trained for 20 epochs, trained with varying values of $\mu$

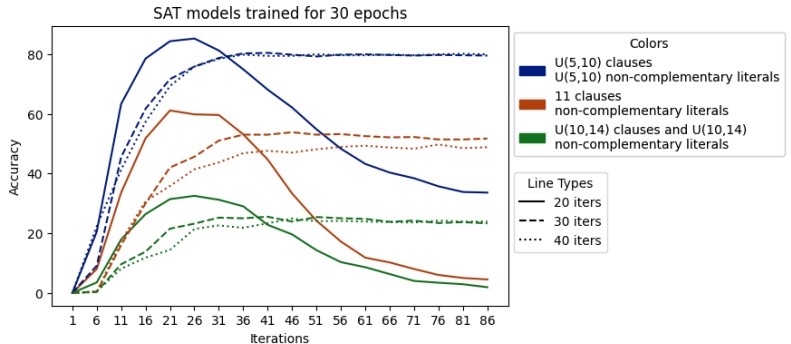

Figure 13: SAT models trained for 30 epochs, trained with varying values of $\mu$

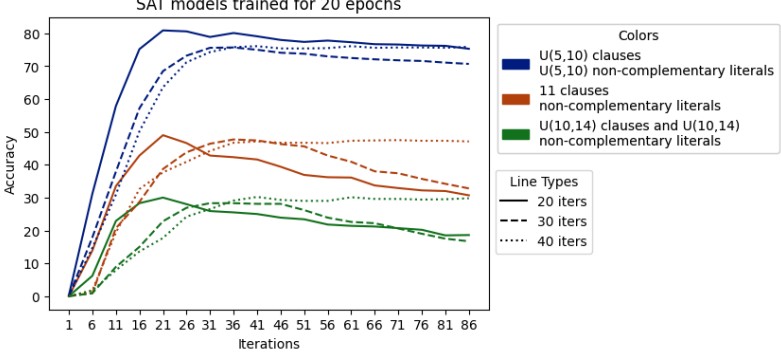

Figure 14: SAT models trained for 20 epochs, trained with varying values of $\mu$

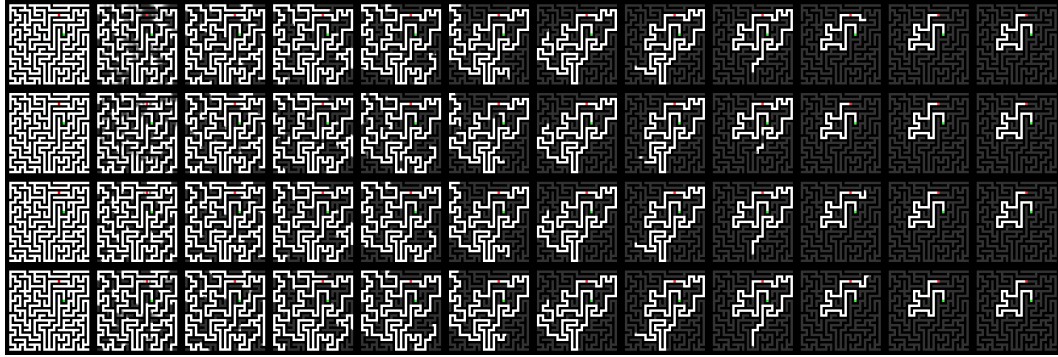

Figure 15: Representative SAT model output every 10 iterations when solving a random training data sample, during testing. The input is shown on the far left.

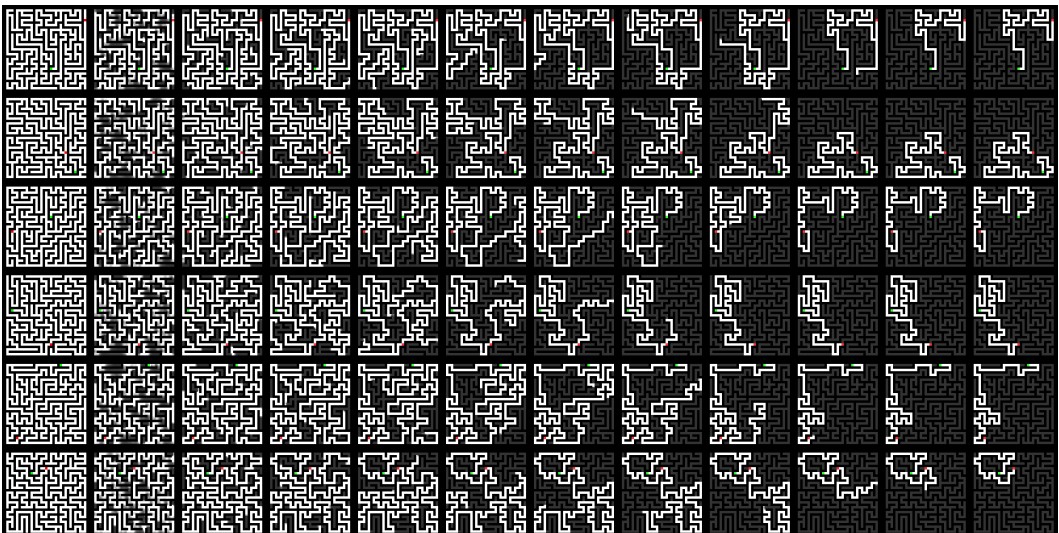

Figure 16: In this figure, we demonstrate that for different models trained with same hyperparameters and $\mu = 30$ always learn the dead-end filling algorithm.

Figure 17: In this figure, we demonstrate that the model trained with $\mu = 30$ has dead-end filling behavior for different mazes.

| Hyperparameter | Mazes | SAT | Prefix Sums |
|---|---|---|---|
| Learning Rate | 0.001 | 0.001 | 0.001 |
| Width | 128 | 512/1024 | 400 |
| Alpha | 0.00 | 0.01 | 1.00 |
| Batch Size | 50 | 5 | 100 |
| Epochs | 50 | 20/30 | 100/300 |

Table 1: The SAT models with $\mu \in \{20, 30\}$ have the larger widths.

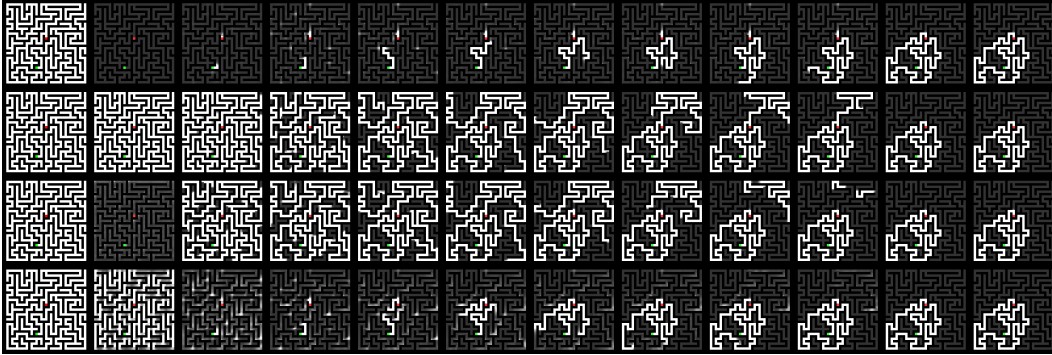

Figure 18: In this figure, we demonstrate that the models trained with $\mu = 20$ for different seeds, switch between the algorithm that mimics dead-end filling and the search-like algorithm.

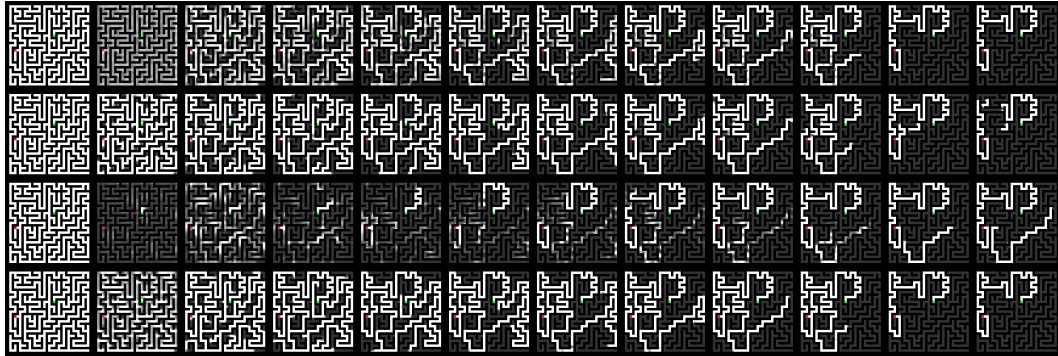

Figure 19: In this figure, we demonstrate that the models trained with $\mu = 15$ for different seeds, switch between the algorithm that mimics dead-end filling and the search-like algorithm.

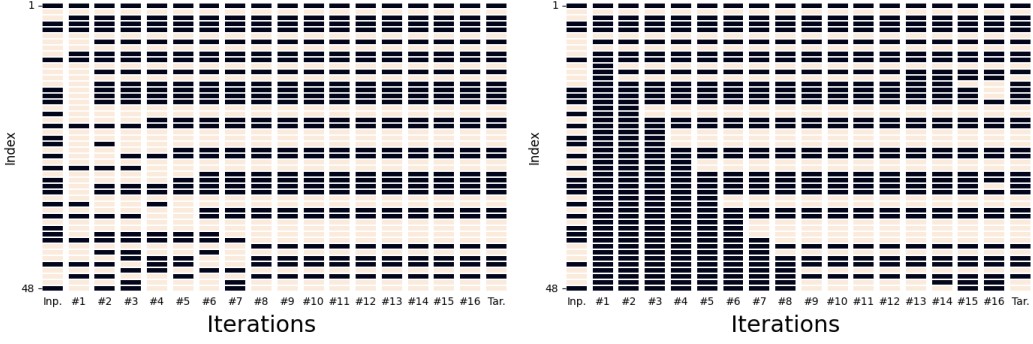

Figure 20: **Left:** Prefix Sums model trained with $\mu = 30$. **Right:** Prefix Sums Prefix Sums model trained with $\mu = 6$

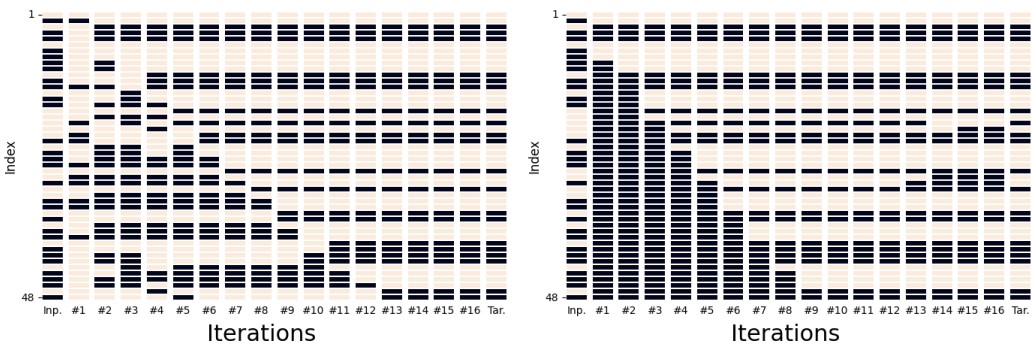

Figure 21: **Left:** Prefix Sums model trained with $\mu = 30$. **Right:** Prefix Sums Prefix Sums model trained with $\mu = 6$

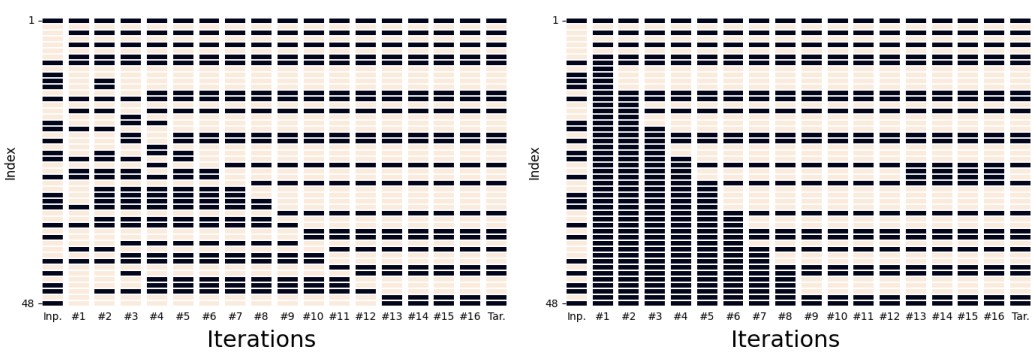

Figure 22: **Left:** Prefix Sums model trained with $\mu = 30$. **Right:** Prefix Sums Prefix Sums model trained with $\mu = 6$

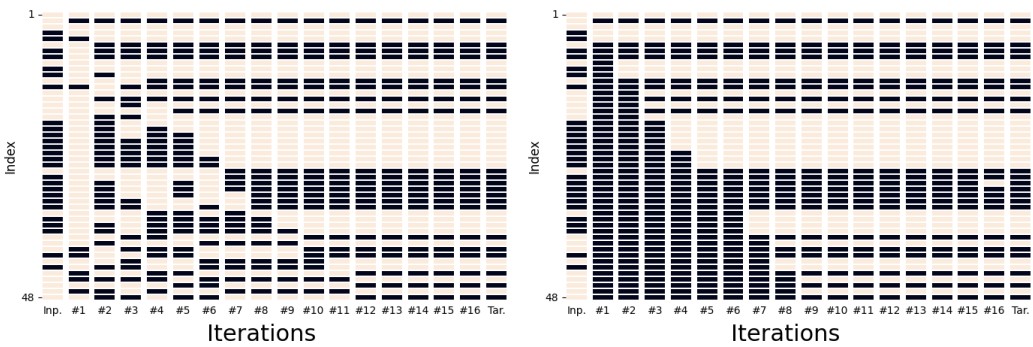

Figure 23: **Left:** Prefix Sums model trained with $\mu = 30$. **Right:** Prefix Sums Prefix Sums model trained with $\mu = 6$

