# OpenReview forum: "Algorithm Design for Learned Algorithms"
_ICLR.cc/2024/Conference — ICLR 2024 Conference Withdrawn Submission_

### Official Review · Reviewer_iu6m · 2023-10-27

**Soundness:** 2 fair
**Presentation:** 3 good
**Contribution:** 2 fair
**Rating:** 3
**Confidence:** 4

**Summary:**

The paper presents an empirical analysis of the behavior of the algorithms learned from data by Deep Thinking (DT) networks. The authors conduct experiments in the domains of computing prefix sums, solving mazes, and constraint satisfiability (SAT). They demonstrate that the hyperparameter μ, which controls the number of iterations during training, influences the behavior of the resulting algorithms and can be used to control the trade-off between accuracy and efficiency.

**Strengths:**

* Studying the behavior and influence of hyperparameter values on the learned algorithms is an interesting domain. The paper demonstrates that the hyperparameter μ, controlling the number of iterations during training, has an impact on the tradeoff between the accuracy and efficiency of the learned algorithms making this in general a work with a potential for significance.
* The authors conducted experiments on various problem domains, including SAT problems, demonstrating that algorithms obtained by DT networks show similar behavior across different problem complexities, which is an interesting finding.
* The paper acknowledges the importance of reproducibility and provides code for experiments and data generation, making it accessible for others to validate the results.
* For most parts, the paper is well-written and easy to follow.

**Weaknesses:**

1. The paper primarily focuses on the empirical study of an existing approach. While this is an interesting direction in general and the finding that the hyperparameter μ can be used to control the accuracy vs. efficiency tradeoff is indeed valuable, in my opinion, I believe the amount of analysis performed is a little bit low and leaves room for deeper analysis.  One could answer many more interesting questions even when only considering the μ hyperparameter. For instance, does it exert varying degrees of influence on different problems? How does it need to be configured to achieve a certain threshold of performance? Moreover, it would be interesting to investigate whether there exists an upper limit beyond which the algorithm's performance does not improve further but merely demands additional computational resources. If such a limit exists, how can it be identified?

2. The experiments mainly focus on relatively simple problems like prefix sums and mazes as test cases. While the experiment on SAT problems shows the adaptability of learned algorithms to NP-Hard problems, the scalability and efficiency of these methods on larger, more complex instances should be thoroughly discussed. Additionally, the number of problems studied could be expanded to achieve a more comprehensive understanding.

3. The paper does only provide very superficial comparisons with existing traditional algorithms for the problems considered. Understanding in more detail how the learned algorithms perform relative to traditional methods would provide more context for their utility. In particular, one could conduct comprehensive quantitative comparisons between the algorithms learned by DT networks and traditional algorithms. Evaluating performance metrics such as execution time, solution quality, and resource usage across various problem instances would help in objectively assessing the strengths and weaknesses of the learned algorithms. Furthermore, it would be possible to perform a qualitative analysis to compare the behavioral patterns of DT network-learned algorithms with those of existing algorithms. Investigating the possibility of using the μ hyperparameter to transition between different existing algorithms would also be interesting.

4. In the abstract and introduction, the authors state that they explore how hyperparameters influence the behavior of the algorithms. However, in practice, only the influence of one single hyperparameter is explored. It would be interesting for the reader how other hyperparameters were set, as well as how they influence the accuracy vs. efficiency tradeoff. Furthermore, it would be interesting to study the importances of hyperparameters, e.g. utilizing fAVONA [Hutter et al., 2014] or Local Parameter Importance [Biedenkapp et al., 2018], as well as the dependencies between hyperparameters. The authors should at least clarify that they only analyze a single hyperparameter.

5. Limitations of the experimental setup and resulting findings are not discussed.

Overall, all of the weaknesses above can be summarized under the umbrella that the amount of analysis performed is not enough for such a paper from my perspective.

[Hutter et al., 2014] Hutter, F., H. Hoos, and K. Leyton-Brown (2014). “An Efficient Approach for Assessing Hyperparameter Importance”. In: Proc. of ICML’14, pp. 754–762.
[Biedenkapp et al., 2018] A. Biedenkapp, J. Marben, M. Lindauer, and F. Hutter. CAVE: Configuration assessment, visualization and evaluation. In R. Battiti, M. Brunato, I. Kotsireas, and P. Pardalos, editors, Proceedings of the International Conference on Learning and Intelligent Optimization (LION), Lecture Notes in Computer Science. Springer, 2018.

**Questions:**

1. I wonder why no standard deviations are shown in the plots. Have the evaluations been run with different random seeds? If so, for how many repetitions?
2. In Section 2.2 the authors claim that algorithms crafted by Deep Thinking networks exhibit a degree of interpretability. But is this interpretability here not rather a consequence of the task at hand than of the Deep Thinking networks? If so, this should be made more explicit.
3. Overall, I wonder how surprising these results really are. In the end μ controls the training time in one way or the other and most iterative learning algorithms get better with a larger training time.

Some minor remarks:

4. Figure 4: Should include a more detailed explanation of what is shown.

5. Figure 6, left part: It probably misses drawn datapoints at the end of the curves as otherwise it is unclear why the curves change in slope at 250 test data bits.

6. Figure 7:
    * Why do some of the test-accuracy curves end before the full number of test-time iterations is reached?
    * What is the difference between “Training with 9 iters” in the title and “Training Iters” in the legend?

7. Figure 10: The color in the legend needs to be fixed.

To consider increasing my rating, I request the following revisions:
* Addressing of at least two of the weaknesses discussed above, i.e. expanding the  empirical study in the direction of two of the following directions:
    * More in-depth analysis of the Influence of the μ hyperparameter
    * More in-depth comparison with existing algorithms
    * Influence of other hyperparameters
* Clarification and/or addressing of the points mentioned in the question section

---

### Official Review · Reviewer_uA8f · 2023-10-31

**Soundness:** 1 poor
**Presentation:** 1 poor
**Contribution:** 1 poor
**Rating:** 1
**Confidence:** 5

**Summary:**

The authors present a method to train interpretable models that learn algorithms from data. The paper is incremental, the presentation of the material is incomplete and certain results are trivial. I propose to reject this paper.

**Strengths:**

None

**Weaknesses:**

This paper is incomplete

The paper lacks rigor and the entirety of section 2 feels hand wavy. I don't understand why I should be convinced that the model learns dead-end filling by observing just three figures. Also, what do to the authors mean by "search algorithm" in this section? The authors claim that the models are extremely interpretable, yet they are forced to use a vague and general name like "search algorithm" to describe what the model is doing?

In Section 3.1 What is the Prefix Sum algorithm? What are you'll trying to solve? What is the model trained on? Prefix sums of what?

Also the result is trivial? For instance. If it takes $\Omega(N)$ to solve a problem, but you give your algorithm only $\O(1)$ steps to solve it, naturally the accuracy of the algorithm will be lower?

The output format of the SAT problem is not described in Section 4.

**Questions:**

None

---

### Official Review · Reviewer_PJg9 · 2023-10-31

**Soundness:** 2 fair
**Presentation:** 2 fair
**Contribution:** 1 poor
**Rating:** 3
**Confidence:** 4

**Summary:**

The paper presents a observational and anecdotal study on the behavior of deep thinking networks (DTN), with a special attention to the effect of the hyperparameter ($\mu$) that controls the number of "reasoning" steps (i.e. repetitions of the recurrent block function) at training time.
The study focuses on three problems: maze solving, prefix (cumulative) sum and SAT.

**Strengths:**

- From a storytelling perspective, the paper is easy to follow
- The paper present a tour of applications for the deep thinking network

**Weaknesses:**

- **Significance.** The work focuses entirely on one specific instantiation of a recurrent architecture, the deep thinking network, rather than studying (or even acknowledging) the existence of a far larger class of recurrent models that share very similar design principles, such as (e.g. deep equilibrium networks [1,2]). Are any of the observations presented in this paper generalizable to these closely related class of models? Is there something specific in the architecture/training of DTN that elicits any of behavior shown in the work? As it stands, the research community targeted by this paper looks rather narrow.
- Even beside this concern, the paper essentially focuses on the number of recurrence steps during training, which has been studied before in various settings (e.g. rate of convergence). These studies should be at the very least mentioned, and the authors should clearly explain what is their work adding to literature.
- Closely related to this, there is only one task (SAT) that has not been addressed before, limiting the originality of the work.
- **Clarity.** Although the storytelling of this work is compelling, the quantity and quality of the details is lacking. This lack of essential details makes it difficult to assess the rigorousness of the work and to move past anecdotal examples. Here's a few precise points on this line:
   - The model/ learning algorithm under analysis is not precisely explained. What is the functional form of the model? What is the learning algorithm? Is the model trained with a fixed-point algorithm, with iterative differentiation, Neumann series, ... ? Are there any constraints in the model output (such as maze path should be connected)? For me this a general requirement, but considering that the DTN is not a widely adopted architecture, it is even more pressing.
  - The single tasks are not well explained. What are the models optimized for in the three settings? What is a formal description of the task? I find especially the one of cumulative sum particularly hard to interpret (how should I read he figures? Is this binary cumulative sum?). Even if the reader can guess, I think the exposition and quality of work would greatly benefit from a more precise introduction of each task, including expected behavior and description of the known algorithms for each task.
  - Caption around figures is lacking. For instance, do Fig 5 to 8 and Fig 10 show behavior of a specific test point, average over test set, ...? Note that for all three presented tasks, in principle it seems to me that it makes sense to define an "inter-example accuracy" (i.e. what's the correct portion of the maze found up to that iteration. If these plots are showing averages (or some other statistics) computed over an entire test set, then I would suggest the authors to report also a measure of spread (e.g. standard deviation, min-max intervals or confidence intervals).
- The work does not disentangle the effect of the architecture and the learning algorithm.
- Claims regarding interpretability are not well supported. For instance, in the maze task, I do not recognize (visually) any particular behavior from the images in Fig 1/2/3. What should the reader looking for? Can the authors present or attempt to sketch a more formal proof of the behavior (dead-end filling) that they claim the model is implementing? This also includes formalizing the 'dead-end filling' "target" algorithm.
- The work misses a discussion of the limitation of the study.

Typo:
Caption fig 4. repeated prefix sums


References
[1] Bai, S., Kolter, J. Z., and Koltun, V. Deep equilibrium models. In Advances in Neural Information Processing Systems, pp. 688–699, 2019.
[2] Grazzi, Riccardo, et al. "On the iteration complexity of hypergradient computation." International Conference on Machine Learning. PMLR, 2020.

**Questions:**

Please see weaknesses.

---

### Official Review · Reviewer_LDuX · 2023-11-01

**Soundness:** 2 fair
**Presentation:** 2 fair
**Contribution:** 2 fair
**Rating:** 5
**Confidence:** 4

**Summary:**

The paper presents an in-depth exploration of Deep Thinking models and their applicability to various algorithmic problems, including polynomial-time solvable problems and NP-Hard problems like the SAT (Satisfiability Testing) problem. The authors introduce a key hyperparameter, μ, which regulates the number of recurrences during training and subsequently affects the reasoning capabilities of the models. Through extensive experiments, the paper demonstrates how varying μ influences the model's speed and accuracy across different problem domains.

The research starts by focusing on problems for which polynomial-time classical algorithms exist, such as prefix sums and mazes. The authors show that they can control the reasoning patterns learned by the models through the manipulation of μ. These findings are then extended to the domain of constraint satisfiability, showcasing that the same level of control over speed and accuracy can be maintained even for NP-Hard problems.

The paper also delves into the reproducibility of the experiments, stating that most of their work can be recreated using code from prior research. Additional training and testing scripts specifically for SAT models are provided in supplementary materials.

Overall, the paper makes several significant contributions:

It extends the applicability of Deep Thinking models from polynomial-time problems to NP-Hard problems.
It introduces and thoroughly investigates the role of the hyperparameter μ in regulating the reasoning capabilities of the models.
It provides empirical evidence to support its claims, including various graphs and tables that visualize the impact of  μ on model performance.
It opens up new avenues for applying neural algorithms to some of the hardest problems in computer science, thereby setting the stage for future research in this area.

**Strengths:**

Originality
Novel Application of Deep Thinking Models: One of the paper's most notable contributions is its exploration of Deep Thinking models in the realm of SAT solving and NP-Hard problems. While Deep Thinking models have been applied to various domains, their applicability to SAT problems is relatively unexplored, making this work original in its approach.

Use of Visual Aids: The inclusion of numerous figures adds to the clarity of the paper, helping to visually represent what might otherwise be complex or abstract concepts.

Broad Applicability: The paper hints at the utility of Deep Thinking models in providing novel reasoning strategies for SAT problems, which could be beneficial for various computational tasks beyond SAT solving.

Contributions to Reproducibility: By making the code and datasets available, the paper adds to the growing body of reproducible research, which is of significant value to the scientific community.

**Weaknesses:**

Lack of Rigorous Comparison with State-of-the-Art Methods
The paper's most glaring weakness is its absence of rigorous comparative analyses with state-of-the-art methods, especially those that solve NP-Hard problems like SAT. Without such a comparison, it is challenging to gauge the true efficacy and novelty of the Deep Thinking Networks (DT-Nets) introduced. The paper mentions existing works like those of Selsam (2019) and Wang et al. (2019a, 2019b, 2021), but does not benchmark the performance of DT-Nets against these or other contemporary models, neither in terms of speed nor accuracy.

Lack of Mathematical Formalism
For a paper that delves into algorithmic reasoning and neural networks' ability to approximate solvers for complex problems, there is a surprising lack of mathematical rigor. While the paper does describe the architecture and training routines, it does not provide the underlying mathematical model or theorems that could prove the DT-Nets' efficacy or limitations. Given the problem's complexity, a robust mathematical foundation would have strengthened the paper's claims.

Unclear Definition and Utility of Hyperparameter μ
The paper discusses the impact of the hyperparameter μ extensively, but it does not sufficiently explain its mathematical or algorithmic significance. While the authors argue that μ allows for a trade-off between speed and accuracy, no empirical evidence supports this claim. There are no clear guidelines or proofs to suggest how μ should be optimally set for different problems or why it is effective.

Limited Variation in Hyperparameters: The paper seems to focus predominantly on the impact of the hyperparameter
μ in the SAT models. Although μ is indeed an important factor, other hyperparameters such as learning rate, batch size, and width have been given limited attention. This lack of comprehensive hyperparameter tuning could limit the generalizability of the results.

Unaddressed Multicollinearity: The paper does not discuss the potential interplay between μ and other hyperparameters. This could introduce multicollinearity into the analysis, which would complicate the interpretation of the results.

Incomplete Analysis on "Overthinking"
The paper introduces the concept of "overthinking" but does not delve deep into its theoretical implications or causes. While it is an interesting observation that the model's accuracy declines after a certain point, a more in-depth analysis or mathematical model to capture this phenomenon would have been beneficial.

Limited Scope of Tested Problems
The paper primarily focuses on prefix sums and maze-solving for experimentation. Although these are computationally interesting problems, they are not sufficiently complex to substantiate the paper’s claims about DT-Nets’ applicability to NP-Hard problems. A broader range of test cases would have provided more robust validation of the proposed approach.

Generalizability and Scalability
The paper claims that the lessons learned from simpler problems like prefix sums and maze-solving can be applied to more complex problems like SAT. However, it does not provide sufficient empirical evidence to substantiate this claim. The scalability of DT-Nets to more complex or larger problems remains unclear.

Ambiguity in Algorithmic Framework: The paper discusses "Deep Thinking models" and their application to SAT problems but fails to provide a clear mathematical formulation or algorithmic outline for these models.

Over-Reliance on Figures: The paper heavily relies on visual representations, such as Figures 10, 11, and 12, for justifying its claims. However, these figures are not accompanied by statistical tests to validate their significance.

Clarity and Precision: Figures like 15, 16, and 17 could benefit from more explicit annotations or captions to aid in understanding what specifically the reader should deduce from them.

Limited Scope: The discussion section is quite generic and does not delve into specific limitations or practical challenges that might arise when applying the Deep Thinking framework to SAT or other NP-Hard problems.

Lack of Novelty: While the paper discusses the potential of Deep Thinking models for SAT problems, it does not articulate how this approach is significantly better or different from existing methods in terms of computational complexity or accuracy.
Lack of Comparative Analysis: The paper does not compare its methodology with state-of-the-art approaches in any metric, making it difficult to assess its contribution objectively.

Silence on Computational Overheads: While the paper discusses the neural algorithmic space for NP-Hard problems, it fails to address the computational overheads involved, which is crucial for comparison with existing algorithms.

**Questions:**

1. Clarification on Hyperparameter μ
Question: Could you elaborate on the theoretical underpinnings that led to the choice of the hyperparameter μ?
Suggestion: A more in-depth discussion on why μ was chosen over other hyperparameters could provide additional rigor to the paper.
2. Comparisons with State-of-the-Art Methods
Question: How do the results compare with existing state-of-the-art SAT solvers or neural network models designed for SAT problems?
Suggestion: Including a comparison with state-of-the-art methods would give the reader a better understanding of where this work stands in relation to existing research.
3. Robustness and Generalizability
Question: Could you discuss the robustness of the proposed method, especially in terms of its performance on edge cases or extremely hard instances of SAT problems?
Suggestion: Consider running additional tests on more challenging problems to demonstrate the robustness of the model.
4. Complexity and Computation Time
Question: What are the computational complexities for the models with different μ values, and how do they affect the model's real-world applicability?
Suggestion: A discussion on computational complexity would add depth and practical relevance to the paper.
5. Reproducibility Concerns
Question: While the paper mentions that the majority of the experiments can be reproduced using prior work, are there any elements of the research that are not easily reproducible?
Suggestion: Clearer guidelines or a dedicated section on the reproducibility of all aspects of the research could be beneficial.
6. Extension to Other NP-Hard Problems
Question: The paper opens up interesting avenues for applying Deep Thinking models to other NP-Hard problems. Have preliminary tests been done on other such problems?
Suggestion: A brief discussion or appendix on preliminary tests in other NP-Hard problems could make the paper more comprehensive.
7. Performance Metrics
Question: Are there other performance metrics, besides accuracy and speed, that could be relevant to evaluate the models?
Suggestion: The inclusion of additional metrics might provide a more rounded evaluation of the proposed approach.
8. Limitations and Future Work
Question: The paper does touch upon future work but doesn't explicitly outline the limitations of the current study. Could you elaborate?
Suggestion: A dedicated section on limitations could lend more balance to the paper and guide future research effectively.